# Greenhouse gas emissions from US irrigation pumping and implications for climate-smart irrigation policy

Avery W. Driscoll [1] ✉, Richard T. Conant[2], Landon T. Marston [3], Eunkyoung Choi[2] & Nathaniel D. Mueller [1,2]

Irrigation reduces crop vulnerability to drought and heat stress and thus is a promising climate change adaptation strategy. However, irrigation also produces greenhouse gas emissions through pump energy use. To assess potential conflicts between adaptive irrigation expansion and agricultural emissions mitigation efforts, we calculated county-level emissions from irrigation energy use in the US using fuel expenditures, prices, and emissions factors. Irrigation pump energy use produced 12.6 million metric tonnes $CO_2$e in the US in 2018 (90% CI: 10.4, 15.0), predominantly attributable to groundwater pumping. Groundwater reliance, irrigated area extent, water demand, fuel choice, and electrical grid emissions intensity drove spatial heterogeneity in emissions. Due to heavy reliance on electrical pumps, projected reductions in electrical grid emissions intensity are estimated to reduce pumping emissions by 46% by 2050, with further reductions possible through pump electrification. Quantification of irrigation-related emissions will enable targeted emissions reduction efforts and climate-smart irrigation expansion.

Food systems produce roughly one-third of global greenhouse gas (GHG) emissions, with confidence intervals of recent estimates ranging from 11 to 22 Gt $CO_2$e $yr^{-1}$ (see refs. 1–3). Rapid reductions in food system GHG emissions will be critical to limiting warming to 1.5° or 2 °C[4]. In tandem with reducing GHG emissions, we must reduce system vulnerability to climate change and increase food production to meet rising demand[5,6]. Irrigation is highly effective at increasing cropland productivity and reducing crop losses associated with drought and heat stress by allowing producers to meet crop water demand regardless of weather and providing both canopy-level and local cooling effects[7–9]. Expansion of irrigation is an increasingly valuable climate adaptation strategy as croplands experience increasing heat stress, precipitation variability, and, in many places, a decrease in total precipitation[10–12]. Irrigated area in the United States continues to expand nationally[13,14] despite regional variability associated with increasing water competition in many regions and ongoing

aridification in the American Southwest[15]. In addition to its adaptive benefits, irrigation also produces GHG emissions through energy use and other sources, potentially conflicting with agricultural sector GHG mitigation goals. However, the magnitude and distribution of irrigation-related emissions are not yet well understood despite the potential for climate-adaptive irrigation expansion to form a reinforcing feedback loop. Climate-smart irrigation policy will require consideration of the greenhouse gas emissions associated with irrigation alongside its adaptive benefits.

Energy is required for extracting groundwater, transporting surface water, and operating pressurized application systems on farms[16]. Off-farm, both infrastructure and energy are required to divert, transport, and store irrigation water[17]. In addition, irrigation increases soil-based emissions of $N_2O$[18], reservoirs required for storage of irrigation water can emit substantial quantities of $CH_4$[19,20], and degassing of supersaturated groundwater emits additional $CO_2$[21]. Notably,

[1]Department of Soil and Crop Sciences, Colorado State University, Fort Collins, CO, USA. [2]Department of Ecosystem Science and Sustainability, Colorado State University, Fort Collins, CO, USA. [3]Department of Civil and Environmental Engineering, Virginia Polytechnic Institute and State University, Blacksburg, VA, USA. ✉e-mail: averywdriscoll@gmail.com

estimates of agricultural sector GHG emissions typically omit emissions associated with energy use, which are instead attributed to the energy sector in accordance with the 2006 IPCC Common Reporting Framework[22]. Within the energy sector, energy use emissions from agriculture, forestry, and fisheries are reported jointly[23], precluding the disentanglement of agricultural GHG emissions specifically. Moreover, these top-down estimates of energy use emissions from agriculture, forestry, and fisheries do not provide insight into specific end uses of energy within the sector. While this classification scheme was developed to avoid double-counting emissions for national reporting, it presents challenges for understanding emissions associated with individual management practices, such as irrigation. Resolving aggregate agricultural GHG emissions to the level of specific management practices, such as those associated with irrigation, will clarify opportunities for agricultural emissions reductions.

While there have been previous efforts to quantify energy use in the US water sector, they have often focused on the public water supply rather than irrigation[17,24-26] despite the fact that water withdrawals for irrigation are approximately three times larger than those for the public supply[27]. Several studies have explicitly evaluated energy use for irrigation pumping, but these studies have generally focused on regional assessments[28], exclusively considered electrical use[29], are not spatially resolved[30], and/or have not estimated associated GHG emissions.

Here, we provide a national-scale, spatially explicit estimate of GHG emissions from energy use for irrigation and assess implications of recent climate policy for reducing these emissions. To do this, we (1) quantified GHG emissions from energy use for irrigation pumping in the US in 2018, (2) assessed drivers of spatial variability in the emissions intensity of irrigation, (3) identified crop-specific contributions to irrigation emissions, and (4) evaluated the projected impacts of the 2022 Inflation Reduction Act on reducing emissions from irrigation pumping. We leveraged state-level data on fuel expenditures for the operation of on-farm irrigation pumps from the US Department of Agriculture Irrigation and Water Management Survey[31] alongside fuel prices (US Energy Information Administration) and emissions factors (US Environmental Protection Agency) to calculate energy- and water source-specific emissions from irrigation pumps. We downscaled these estimates to the county level based on the volume of water withdrawn for crop irrigation in each county[27], adjusted for groundwater depth[32]. We then estimated irrigation energy use emissions for 12 individual major irrigated crops at the county level, using data on crop-specific irrigated area[33] and crop water demand[31] adjusted for aridity[34]. Finally, we projected the impacts of grid decarbonization[35] and irrigation pump electrification on future irrigation emissions.

## Results and discussion

### Groundwater use and electrical pumps dominate total emissions

In 2018, energy use for on-farm irrigation pumps in the US produced ~12.64 million metric tonnes (MMT) $CO_2e$ (Fig. 1; 90% CI: 10.44, 15.05 MMT $CO_2e$), a share equivalent to 16% of the total energy use emissions attributed to the agriculture, forestry, and fisheries sector in the US[23]. This corresponds to an estimated 156 PJ of total energy use for on-farm irrigation pumping. Previous studies have estimated on-farm irrigation pump energy use at 158 PJ nationally[30] and 136 PJ for electricity use in the Western USA[29], in close agreement with our estimates. Supplementary Table S1 provides energy and emissions intensity estimates from selected regional and international studies, demonstrating substantial variability in estimated intensities. On a per-hectare basis, for instance, energy intensity estimates have ranged from 6687 MJ ha$^{-1}$ (our study) up to 43,412 MJ ha$^{-1}$ (from a study of groundwater irrigation in Pakistan), while emissions intensity estimates have varied from 0.54 tonnes $CO_2e$ ha$^{-1}$ (our study) up to 1.27 tonnes $CO_2e$ ha$^{-1}$ (from a study of groundwater irrigation in India).

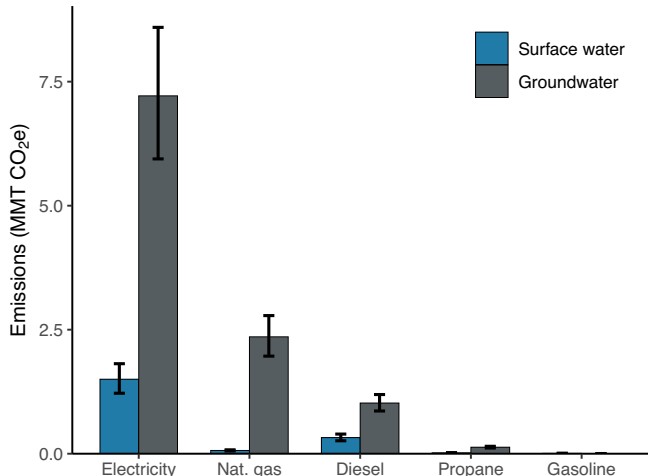

**Fig. 1 | National greenhouse gas emissions from irrigation pumping by fuel and water source.** Total greenhouse gas emissions (million metric tonnes $CO_2e$) from energy use for on-farm irrigation pumping in 2018 in the United States by pump energy source for surface water (blue bars) and groundwater (gray bars). Error bars indicate 90% confidence intervals.

Previous studies have often focused on groundwater pumping, which is associated with higher energy use and subsequently higher emissions. In addition, most of these studies rely on a bottom-up approach, using information about pumping depths, volumes, and efficiencies to calculate the theoretical pump energy requirements. Further comparisons between bottom-up and top-down approaches (such as that used in this manuscript) would be useful to increase confidence in irrigation energy use estimates.

Although groundwater accounted for only 48.5% of irrigation water withdrawals[27], groundwater pumping contributed 85.0% of the total emissions, or 10.73 MMT $CO_2e$ (90% CI: 8.86, 12.73 MMT $CO_2e$). Groundwater depth is a primary determinant of the energy intensity of groundwater extraction for irrigation, so emissions from groundwater utilization will increase as aquifer levels decline in areas where rates of extraction exceed rates of recharge, as is the case over large portions of the High Plains Aquifer[28,36,37]. Notably, the average rate of emissions per m$^3$ of groundwater used for irrigation was over five times larger than the rate of emissions per m$^3$ of surface water used for irrigation (138 g $CO_2e$ m$^{-3}$ for groundwater vs. 28 g $CO_2e$ m$^{-3}$ for surface water). Operations utilizing groundwater typically rely on pressurized irrigation systems with higher water use efficiencies (such as sprinkler, drip, or micro) due to the relatively high cost of water extraction. In contrast, some operations utilizing surface water in conjunction with gravity-fed irrigation systems (such as furrows or flooding) do not require any pumping. Although utilization of gravity systems has been declining in the interest of water conservation[14], gravity systems were still used on 36% of irrigated area as of 2015[27].

Electricity use dominated pumping emissions relative to other energy sources, accounting for 68.9% of total emissions (Fig. 1; 8.72 MMT $CO_2e$; 90% CI: 7.16, 10.41 MMT $CO_2e$) and approximately the same proportion of pumped irrigated area (67.8%). Natural gas accounted for 19.2% of emissions (2.42 MMT $CO_2e$; 90% CI: 2.03, 2.86 MMT $CO_2e$) but only 7.5% of pumped irrigated area. In contrast, diesel fuel was utilized on 22.4% of the area but accounted for only 10.6% of total emissions. Propane (1.2% of emissions and 1.9% of area) and gasoline (0.08% of emissions and 0.2% of area) are no longer widely utilized for irrigation pumping in the US. Natural gas and electrical pumps were associated with much higher energy demand per irrigated hectare than pumps using other fuels after accounting for fuel-specific differences pump efficiency (Supplementary Fig. S1A). This finding indicates that operations with higher pumping energy

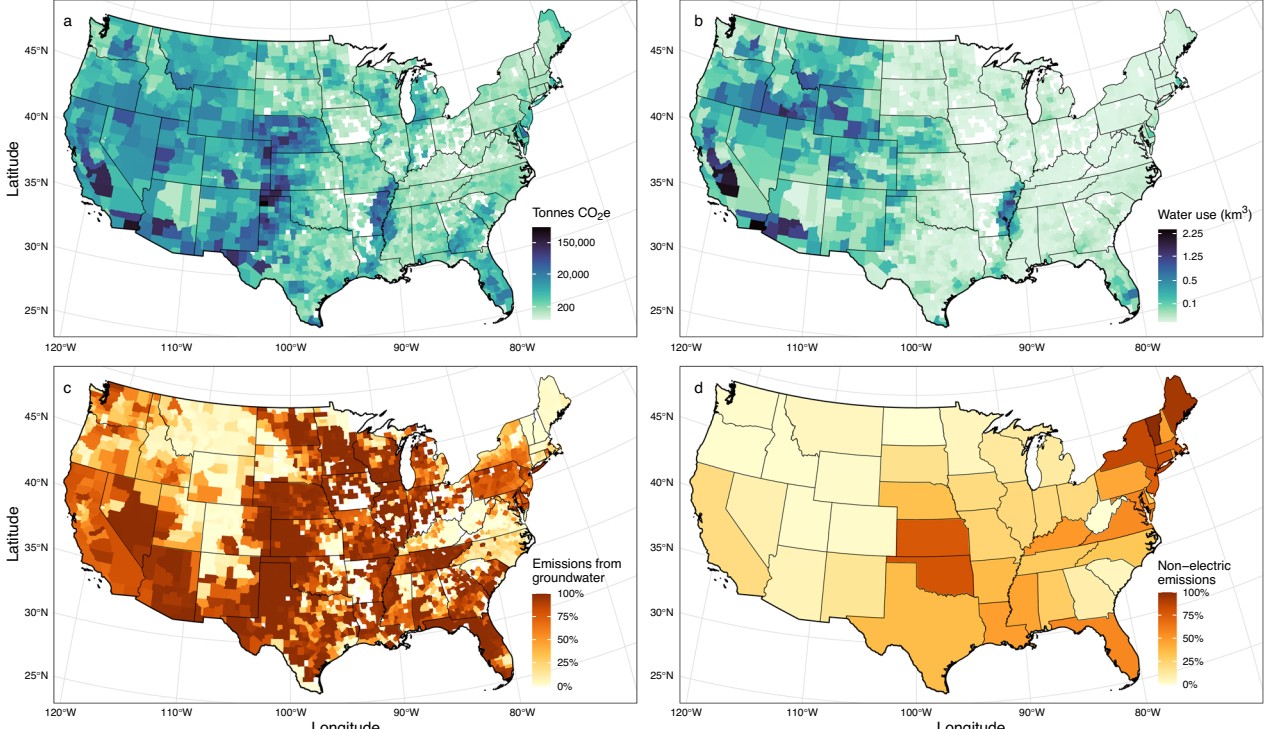

**Fig. 2 | County-level map of greenhouse gas emissions from irrigation pumping.** The distribution of (**a**) total greenhouse gas emissions (tonnes CO₂e) from energy use for on-farm irrigation pumping in 2018 in the United States, and several key drivers of emissions, including: **b** total water demand for crop irrigation (km³ per year), **c** the percentage of total emissions attributable to groundwater use rather than surface water use, and **d** the percentage of emissions attributable to natural gas, diesel, gasoline, or propane pumps rather than electrical pumps. **a, b** The color scale is square root transformed for improved visibility. **a–c** Counties that have no emissions are in white.

demands, perhaps due to high groundwater reliance, high water demand, or deeper groundwater levels, were more likely to utilize natural gas or electrical pumps. After adjusting for efficiency, we found that the average emissions factor for electric pumps was 11 g CO₂e MJ⁻¹, much lower than natural gas (24 g CO₂e MJ⁻¹) or other fuels (Supplementary Fig. S1B; range 23–30 g CO₂e MJ⁻¹). Solar-powered pumps were utilized on only 0.3% of pumped irrigated area in 2018 (60,854 ha), however, interest in solar pumping has accelerated in recent years as economic feasibility has improved[38–40]. Although deployment remains limited, the area utilizing solar pumps increased more than fivefold from 2013 (11,373 ha), indicating that there is substantial momentum towards expansion[41]. Improvements in irrigation water use efficiency have previously been shown to actually increase total water use[28,42,43]. Because increased pump energy use efficiency may similarly increase water use due to reduced pumping costs, safeguards against overextraction of water resources should be considered alongside incentives for the adoption of solar and electrical pumps.

## High spatial variability in county-level emissions

Irrigation pumping emissions varied considerably across the country (Fig. 2a). The three states with the highest emissions (Texas, Nebraska, and California) accounted for 46.0% of national GHG emissions and 39.4% of irrigated croplands. In contrast, the 24 states with the lowest emissions, which were concentrated in the Northeast and the Upper Midwest, accounted for only 3.3% of total emissions and 6.7% of total irrigated crop area. State-level emissions estimates and emissions intensities per irrigated hectare and per m³ of irrigation water used are shown in Supplementary Fig. S2. Area-based average emissions intensities ranged from 143 kg CO₂e ha⁻¹ (Idaho) to 1929 kg CO₂e ha⁻¹ (New Mexico), and were generally highest in arid states, including Texas (1710 kg CO₂e ha⁻¹), Arizona (1576 kg CO₂e ha⁻¹), and Oklahoma

(1462 kg CO₂e ha⁻¹). In the four states with the highest emissions intensities, mean water demand (2460 m³ ha⁻¹), the share of emissions from non-electric pumps (35.8%), and groundwater depths within irrigated areas (37.8 m) all exceeded the national median values (568 m³ ha⁻¹, 26.2%, and 10.6 m, respectively).

At the county level, irrigation pumping emissions were significantly positively associated with irrigated area, the volume of irrigation water use (Fig. 2b), relative reliance on groundwater (Fig. 2c), and groundwater depth ($P < 0.001$ for all; coefficient estimates are provided in Supplementary Fig. S3). Average pump fuel efficiency, which is much higher for electrical pumps (88%) than propane, gasoline, natural gas, or diesel pumps (25%, 23%, 21%, and 31%, respectively)[44], was also strongly negatively associated with emissions (Fig. 2d and Supplementary Figs. S3B and S4). Areas of high GHG emissions coincided with several distinctive agricultural regions, including the High Plains Aquifer, the Mississippi Delta, the Central Valley in California, and the Gila and Imperial Valleys in southern Arizona and California. The 237 counties located over the High Plains Aquifer had a particularly outsized contribution to national emissions, accounting for 44.7% of irrigation energy use emissions (5.63 MMT CO₂e; 90% CI: 4.82, 6.49 MMT CO₂e) despite containing only 27.6% of all irrigated area. Our estimate of emissions intensity in this region (248 g CO₂e per m³ of water) agrees quite well with a bottom-up estimate produced by a previous local study of the Kansas High Plains Aquifer (231 g CO₂e per m³)[28] (Supplementary Table S1). For the Kansas High Plains Aquifer specifically, we calculated an emissions rate of 275 g CO₂e per m³. The region is heavily reliant on groundwater, which accounted for 83.1% of its irrigation water use and 97.2% of pumping emissions. Moreover, its groundwater stores are substantially deeper than the national median over irrigated areas (24.7 m versus 10.6 m), and Kansas and Oklahoma in particular had a very large share of emissions coming from natural

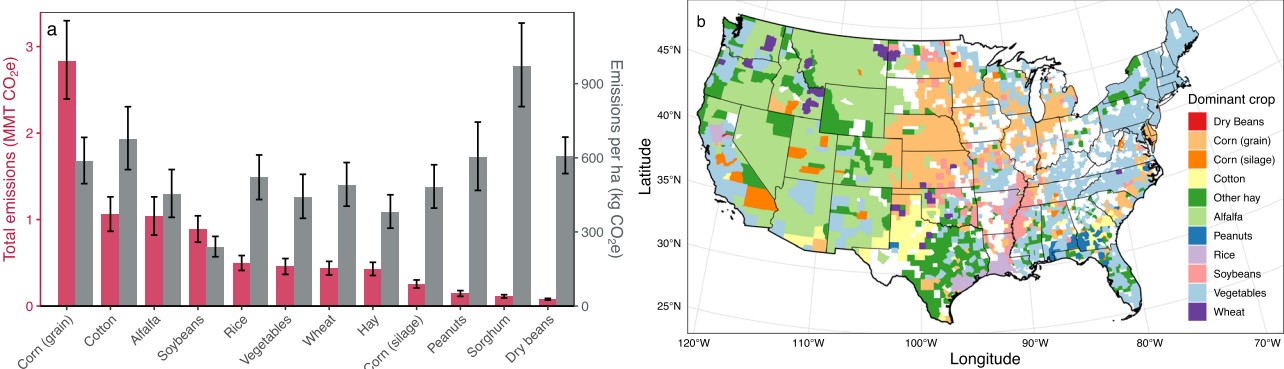

**Fig. 3 | Crop-specific greenhouse gas emissions from irrigation pumping. a** Bar plot showing total greenhouse gas emissions (red bars) and emissions per hectare (gray bars) associated with on-farm irrigation pumping for 12 of the most widespread irrigated crops in the US. **b** Map showing the crop associated with the largest proportion of irrigation pumping emissions in each county. Counties with no pumped irrigated area are shown in white, and counties without crop-specific emissions estimates due to data withholding for confidentiality are shown in gray. Error bars indicate 90% confidence intervals.

gas pumps (72.3%), contributing to the region's low average pump fuel efficiency (Supplementary Fig. S4).

The Colorado River Basin serves as a useful counterpoint. Despite its more arid climate, it has a much lower emissions intensity than the High Plains Aquifer region (670 kg $CO_2$e ha$^{-1}$ versus 1089 kg $CO_2$e ha$^{-1}$) because of differences in water and fuel sources. The Colorado River Basin relies heavily on surface water (72.4% of total withdrawals) instead of groundwater, and it is heavily electrified, with 87.9% of emissions on average coming from electric pumps across the seven Colorado River Basin states. Thus, the share of national pumping emissions that occur in the Colorado River Basin (8.9%) is roughly proportionate to its share of irrigated area (7.4%). Similarly, high rates of electric pump adoption, low relative reliance on groundwater, and low electrical grid emissions intensity interact to produce relatively low emissions intensity in the Northwestern US despite high irrigation water use.

## Crop-specific pumping emissions driven by geography and extent

We allocated emissions to 12 major irrigated crops based on county-level emissions rates per m³ water withdrawn, crop-specific rates of irrigation water application, and crop-specific irrigated area, accounting for 74.3% of the total irrigated area (Fig. 3). This analysis required integration of water use data from both the USDA and the USGS, which have discrepancies in methodology and subsequent total estimates of irrigation water use, likely resulting in absolute estimates of crop-specific water use that are too low. However, the relative contribution of each crop, the spatial distribution of crop-specific estimates, and all non-crop-specific results are unaffected by the discrepancies between the USDA and USGS data. Further implications are detailed in the Supplementary Discussion. We found that irrigation pumping for corn for grain produced the most total emissions by a large margin (2.82 MMT $CO_2$e, 90% CI: 2.40, 3.30 MMT $CO_2$e), in part due to its large irrigated extent (4.8 Mha; crop areas shown in Supplementary Fig. S5A). The high-emissions-intensity High Plains Aquifer region accounted for 76.9% of the crop's total emissions (Fig. 3b and Supplementary Fig. S6), although irrigated corn for grain was also present throughout much of the Midwest, the Southeast, and the Mississippi Delta. Despite its high total emissions, corn for grain had a lower emissions intensity per irrigated hectare (585 kg $CO_2$e ha$^{-1}$) than several other crops, in part because of its relatively low average water use (2653 m³ ha$^{-1}$; crop water application rates shown in Supplementary Fig. S5B).

In contrast to corn for grain, sorghum and cotton had particularly high-emissions intensities (Fig. 3a; 970 and 675 kg $CO_2$e ha$^{-1}$, respectively) due to their geographic distribution (Fig. 3b and Supplementary Fig. S7). Sorghum was grown almost exclusively in the Texas and Oklahoma panhandles, western Kansas, and eastern Colorado, where groundwater reliance is high and electrical pump adoption is relatively low. Despite its high emissions intensity, the total emissions attributable to sorghum were low because irrigated sorghum is not widespread (0.1 Mha). Cotton emissions were also concentrated in highly arid areas with high groundwater reliance, including northern Texas, California's Central and Imperial Valleys, southern Arizona, and southeastern New Mexico, contributing to its high-emissions intensity. Due to the combination of its emissions intensity and large extent (1.6 Mha), the irrigation of cotton produced the second-highest emissions after corn.

Soybeans had the lowest emissions intensity of all assessed crops (239 kg $CO_2$e ha$^{-1}$), due to low average water requirements (1935 m³ ha$^{-1}$) and greater spatial concentration in the Mississippi Delta, where shallow groundwater and moderate pump fuel efficiency result in somewhat lower energy requirements. Alfalfa and other hay crops, which tend to have very high water requirements (6965 and 5103 m³ ha$^{-1}$, respectively), were responsible for the largest share of irrigation emissions in many counties in the Western US. Taken together, they were associated with 1.46 MMT $CO_2$e (90% CI: 1.17, 1.77 MMT $CO_2$e), though emissions were highly concentrated in just a few counties. For example, Imperial County, CA and Maricopa County, AZ alone were responsible for 15% of total alfalfa irrigation emissions. Despite their high water demands, the average emissions intensities for alfalfa and other hay irrigation were relatively low (451 kg $CO_2$e ha$^{-1}$ and 380 kg $CO_2$e ha$^{-1}$, respectively), influenced by the large area across the Northwestern US where groundwater reliance is low, electric pump adoption is high, and the emissions intensity of the electrical grid is low.

## Strong potential for pumping decarbonization under the Inflation Reduction Act

Given that 69% of emissions were attributable to electricity use for pumping, reductions in the emissions intensity of the electrical grid will substantially reduce emissions from irrigation pumping without additional changes in producer behavior (Fig. 4). To project future irrigation pumping emissions assuming constant energy demand for irrigation pumping, we leveraged state-level projections of changes in electrical grid emissions factors through 2050 under three policy scenarios: (1) current policy, including the projected impacts of the Inflation Reduction Act (IRA), (2) a counterfactual scenario in which the IRA had not been passed, and (3) a scenario in which two key IRA renewable energy tax credits, the Production Tax Credit (PTC) and the

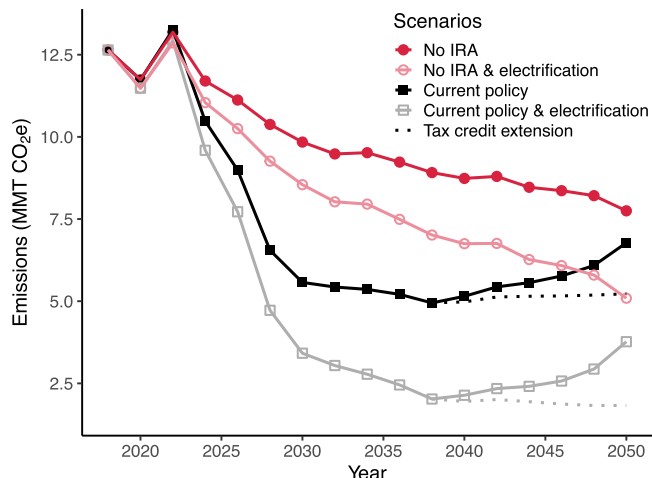

**Fig. 4 | Effects of grid decarbonization on future irrigation pumping emissions.** Projected greenhouse gas emissions from irrigation pumping from 2018 to 2050 associated with changes in the fuel mix for electricity generation under three policy scenarios: current policy, including the Inflation Reduction Act (IRA; black and gray squares); no IRA (red circles); and an extension of key tax credits expanded by the IRA (dashed lines). For each of these, we additionally incorporated electrification of non-electric irrigation pumps at a rate of 5% of remaining non-electric energy use per year (open symbols).

Investment Tax Credit (ITC), remain in place[35]. Under current policy, annual pumping emissions are projected to decrease by 46% by 2050 relative to 2018 levels without any on-farm changes in energy source or demand. Due to the phase-out of the PTC and ITC that is scheduled to begin in 2032, emissions are projected to reach a minimum in 2038 and then begin to increase again through 2050. Keeping these tax credits in place would reduce annual emissions by 59% in 2050 relative to 2018. Without the IRA, we project only a 39% emissions reduction by 2050.

Although projected grid improvements will substantially reduce irrigation pumping emissions, full decarbonization will require both a net-zero-emission electrical grid and a transition from natural gas, diesel, propane, and gasoline to electricity or other zero-emission energy sources. For each policy scenario, we additionally modeled the effects of fuel-switching from natural gas, diesel, propane, and gasoline to electricity at a rate of 5% of remaining non-electric energy use per year (Fig. 4, open symbols). Because barriers such as electrical grid connectivity may preclude electrification in some irrigated areas, our electrification scenario implies that ~8% of 2018 non-electric fuel use will not be electrified by 2050. This rate does not attempt to reflect historical trends, but rather serves as a benchmark to exemplify decarbonization potential associated with electrification. Pump electrification coupled with current policy could decrease annual pumping emissions by 70% in 2050 relative to 2018, and by 86% if the PTC and ITC are extended. Compared to the scenario with no IRA or electrification, current policy coupled with the tax credit extension and pump electrification would reduce irrigation energy use emissions by an additional 5.92 MMT $CO_2e$ annually by 2050, to only 1.83 MMT $CO_2e$ yr$^{-1}$. Electrification will additionally reduce total energy demand due to increased efficiency, helping to offset anticipated increases in energy demand associated with groundwater depletion and irrigation expansion. Under our electrification scenario, total pumping energy demand would fall by 29.9% by 2050 relative to 2018 (saving ~46.8 PJ yr$^{-1}$).

Further work is needed to understand and reduce infrastructural and economic barriers to pump electrification. Electrification requires an up-front investment in a new pumping system, a reliable connection to the electrical grid, and potentially higher operational costs depending on local energy prices. Strategic incentives for electric pump adoption could offset cost-related barriers and help to accelerate pump electrification. For areas where grid connectivity is limiting, expansion of solar pump usage may be needed to fully eliminate pumping emissions. In addition, scenario analyses that integrate the effects of pump electrification and grid decarbonization with variables that will influence pump energy demand would be useful to clarify future projections of irrigation-related emissions. For instance, declining groundwater levels, migration of irrigation and crop types, changes to surface versus groundwater reliance, and changes to irrigation water demand due to climate change will all affect total pump energy requirements.

Importantly, reducing emissions from irrigation will require more than simply decarbonization of energy use for on-farm pumping. Several additional emissions sources contribute to the total GHG footprint of irrigation, such as elevated $N_2O$ emissions due to increased soil moisture, degassing of groundwater supersaturated in $CO_2$ and $N_2O$, energy use for off-farm pumping, and $CH_4$ production from reservoirs used for irrigation. Thus, the total GHG emissions impact of irrigation is higher than the values presented here, which include only emissions from on-farm pumping. $CO_2$ degassing from groundwater depletion for all end uses has been estimated at 1.7 MMT $CO_2e$ annually[21], while field-scale studies have identified increases in $N_2O$ emissions of up to 170% under irrigated vs. rainfed conditions[45]. We expect spatial patterns of these additional emissions to vary, exacerbating total irrigation-related emissions in areas with extensive groundwater depletion (due to groundwater $CO_2$ degassing), in arid areas (due to larger impacts on soil moisture and subsequently $N_2O$ emissions), and in areas with heavy reliance on imported water (due to pumping requirements and extensive reservoir storage). Mitigation of emissions from these alternative sources is likely to be more complex than mitigation of energy use emissions; a comprehensive quantification will help guide GHG reduction efforts.

As the irrigated area continues to expand in the US and globally, we suggest that the carbon costs of irrigation expansion be considered in conjunction with the sustainability of water withdrawals and anticipated benefits to agricultural yields and system stability. Here, we identified that energy use for irrigation pumping contributes substantially to agricultural sector GHG emissions in the US, and that irrigation pumping emissions are highly spatially heterogeneous. Irrigation expansion is likely to be less emissions-intensive in areas with low groundwater dependence, shallow water tables, lower supplemental water requirements, high rates of electrical pump adoption, and cleaner electrical grids. Common frameworks for national-scale GHG accounting are limited with respect to their ability to resolve emissions to specific subsectors and management practices, and the emissions impacts of energy use for irrigation had not previously been evaluated at the national scale. However, this degree of resolution is useful for identifying potential adaptation-mediated feedback loops and for targeting emissions reduction efforts. Ultimately, a combination of grid decarbonization and policies incentivizing electric and solar pump adoption will be needed to fully eliminate emissions from energy use for irrigation pumping.

## Methods

Emissions from energy use for irrigation were calculated according to Eq. (1), where $E_{f,w,s}$ denotes emissions (tonnes $CO_2e$) for each fuel ($f$), water source ($w$), and state ($s$). $D_{f,w,s}$ denotes per-acre expenditures on fuels for irrigation pumps, $A_{f,w,s}$ denotes the acres irrigated, $P_{f,s}$ denotes the price of the fuel, and $F_{f,s}$ denotes the emissions factor for the fuel. Emissions for each fuel and water source were summed to determine total state-level emissions. State-level rates of emissions per irrigated hectare were calculated by dividing the total emissions by the irrigated hectares in each state. All data used for the calculation of energy use emissions are publicly available, and data sources are described in

Supplementary Table S2. This approach broadly follows that of Sowby and Dicataldo[30], who first presented national-scale estimates of irrigation energy use by integrating data on pump fuel expenditures with fuel prices.

$$E_{f,w,s} = \frac{D_{f,w,s} \times A_{f,w,s}}{P_{f,s}} \times F_{f,s} \qquad (1)$$

### Fuel expenditures

State-level data on per-acre fuel expenditures ($D_{f,w,s}$) for the operation of on-farm irrigation pumps were acquired from the 2018 USDA Irrigation and Water Management Survey (IWMS), a Census of Agriculture supplement that collects detailed information from irrigators[31]. Although survey data are subject to limitations, such as incomplete response rates and potential misreporting by respondents, they are the best available data at the national scale. In addition, the IWMS provides coefficients of variation alongside the data, which were incorporated into the uncertainty analysis. Because irrigators do not directly report energy use, we rely on integration of energy expenditures and energy prices as a proxy for energy use. Per-acre expenditures and irrigated area ($A_{f,w,s}$) are reported separately for each fuel type (electricity, natural gas, gasoline, diesel, and propane) and water source (surface water and groundwater). Total expenditures for on-farm irrigation pumping were calculated for each state by multiplying the per-acre expenditure by the irrigated area for each fuel and water source combination. Some state-level data on expenditures and/or associated area were not available due to data withholding for confidentiality. By leveraging discrepancies between the state-level USDA data and national totals (as detailed in Supplemental Methods), we identified that these omitted data represented a small percentage of both the total area (0.48%) and the total emissions (0.42%). Emissions estimated from these data are included in the reported national totals (Figs. 1 and 4), but not in any spatially explicit or crop-specific results.

### Fuel prices

To calculate the quantity of fuel used, fuel expenditures were divided by 2018 fuel prices ($P_{f,s}$) acquired from the US Energy Information Administration (EIA) and aggregated to the state level. Price data are reported by the EIA at different spatial and temporal resolutions for each fuel; details are provided in the supplementary methods. The EIA data for diesel and gasoline include federal and state excise taxes that are applicable to on-road users but not agricultural users. To better reflect off-road diesel and gasoline prices, federal and state taxes were subtracted from the on-road prices. For electricity prices, we used EIA prices for industrial users. To our knowledge, these are the best available data, but we acknowledge that there is substantial variation in electrical pricing between individual utilities. Thus, the prices paid by irrigators may diverge from the EIA estimates, and the development of energy price estimates specific to agricultural users would be useful for future analyses. Because electricity and natural gas are purchased at the time of use and cannot be stored, we included only prices from months during which the average minimum temperature was greater than 0 °C. To calculate the average minimum temperature for this purpose, daily data on 2018 minimum temperatures (PRISM Climate Group, 4-km resolution) were aggregated to monthly, county-scale values by weighting each PRISM grid cell by the proportion of area classified as cropland by the GFSAD30NACE product from NASA EarthData (30-m resolution). Temperatures were then weighted by the extent of irrigated, harvested cropland area in each county based on the 2017 Census of Agriculture and averaged to the state level.

### Emissions factors

Emissions factors ($F_{f,s}$) for the stationary combustion of natural gas, distillate fuel oil no. 2 (diesel), motor gasoline, and propane were taken from the Environmental Protection Agency (EPA) Greenhouse Gas Emissions Factor Hub and converted to $CO_2$e-based on the 100-year global warming potential for $CH_4$ and $N_2O$. Electricity emissions factors are spatially variable, as they depend on the fuel mix used for electricity generation. We used state-level emissions factors from the EPA Emissions and Generation Resource Integrated Database (eGRID), which are based on in-state electricity generation. These emissions factors do not account for the transmission of electricity across state borders, and irrigators near state boundaries may have out-of-state electricity providers.

### Uncertainty analysis

We used a Markov Chain Monte Carlo approach to capture uncertainty inherent in the IWMS irrigated area data, the IWMS fuel expenditure data, and the EIA fuel price data. Truncated normal distributions with a lower bound at zero were generated for each variable based on the reported coefficient of variation from the IWMS data and the standard deviation of the temporal variability in the fuel price data. We calculated emissions estimates 10,000 times by randomly resampling with replacement from each generated distribution. The 5th and 95th percentile emissions estimates were recorded for each state, fuel, and water source combination and used for the construction of 90% confidence intervals throughout.

### County-level downscaling

State-level emissions estimates were downscaled to the county level ($E_c$) based on the volume ($V$) of surface ($a$) and ground ($g$) water withdrawn for agricultural irrigation in each county, as reported by the US Geological Survey (USGS) in the Estimated Use of Water in the United States County-Level Data for 2015[27], according to Eq. (2). USGS estimates of surface water withdrawals were adjusted for national average conveyance losses of 15.9%, as estimated by the 2019 USDA Survey of Irrigation Organizations[46]. For 13 states, the USGS estimates of crop water withdrawals also include golf course irrigation. Adjustments to exclude golf course irrigation are detailed in the Supplementary Methods.

 Source-specific, county-level water withdrawals for agricultural irrigation were summed to the state level, and the source-specific rate of emissions per cubic kilometer of water withdrawn per year was calculated for each state and water source by dividing the mean, 5th percentile, and 95th percentile emissions estimates by the volume of water withdrawn. County-level surface water emissions were calculated by multiplying surface water emissions rates by the surface water withdrawals ($V_{c,a}$). To account for variable groundwater depths within states, we scaled county-level groundwater emissions rates by groundwater depth. We first calculated county average water table depths using modeled estimates (1 km resolution) of year-round average water table depth between 2004 and 2014, updated in 2020 by Fan et al.[32]. We then calculated the weighted average state-level water table depth based on the volume of water withdrawn in each county and calculated county-level scaling factors as the county average water table depth ($D_c$) divided by the state average water table depth ($D_s$). County-level groundwater emissions were calculated by multiplying state-level groundwater emissions rates by this scaling factor and county-level groundwater withdrawals ($V_{c,g}$). County-level, water source-specific emissions estimates and irrigation water volumes are provided in Supplementary Data 1.

$$E_c = \left( \frac{E_{s,a}}{V_{s,a}} \times V_{c,a} \right) + \left( \frac{E_{s,g}}{V_{s,g}} \times \frac{D_c}{D_s} \times V_{c,g} \right) \qquad (2)$$

### Emissions allocation to major field crops

We estimated the relative contribution of 12 major field crops ($t$) to emissions in each county based on county-level emissions rates and

crop-specific data on the volume of water applied per acre ($R_{s,t}$) from the 2018 IWMS data. Crops included corn for grain, corn for silage, soybeans, wheat, alfalfa hay and haylage, other hay and haylage, rice, cotton, peanuts, dry beans, sorghum for grain, and vegetable totals, and available area data for these crops accounted for 74.3% of irrigated area. Hay and haylage data for both alfalfa and other hay include only harvested area, not pastured area. The selection of crops was based on data availability and the extent of irrigated area. Treatment of missing water application rate data for these crops is described in the Supplementary Methods and affected only 0.1% of total irrigated area. Irrigated pasture was omitted due to a lack of data, as were orchards, barley, and sugar beets. County- and crop-specific emissions ($E_{c,t}$) were estimated according to Eq. (3), where $E_c$ refers to the total irrigation emissions in the county (including the 5th and 95th percentile estimates), $V_c$ refers to the total volume of water withdrawn for irrigation in the county, and $A_{c,t}$ refers to the county-level irrigated area for each crop.

$R_{c,t}$ refers to the scaled, county-level rate of water applied for each crop type ($t$), which was calculated by scaling $R_{s,t}$ by precipitation (P) minus reference evapotranspiration ($ET_o$) for months in 2018 with an average minimum temperature above 0 °C. All meteorological and other input data were cropland-weighted according to the procedures described in the fuel price section. Daily $ET_o$ for 2018 was calculated according to the FAO Penman-Monteith equation[47] using meteorological data from PRISM and AgERA5[34] and elevation data from GMTED2010[48]; calculation details are provided in the Supplementary Methods. We fit a linear regression with a log link between the state-level P-$ET_o$ and $R_{s,t}$, which was then used to estimate county-level rates of water applied based on county-level P-$ET_o$. $R_{c,t}$ was multiplied by the irrigated area for each crop and county ($A_{c,t}$) to calculate the total volume of water applied. Finally, these values were reconciled with the state-level census data by multiplying the total volume by the ratio of the state-level volume over the state sums of the county-level volume.

$$E_{c,t} = \frac{E_c}{V_c} \times R_{c,t} \times A_{c,t} \times \frac{R_{s,t} \times A_{s,t}}{\sum(R_{c,t} \times A_{c,t})} \qquad (3)$$

### Potential impacts of grid decarbonization and pump electrification on emissions

We leveraged state-level projections of electrical grid emissions factors from 2022 to 2050 from the National Renewable Energy Laboratory's (NREL) 2022 Standard Scenarios Report[35] to project potential declines in irrigation energy use emissions associated with changes to the fuel mix for electricity generation. We selected three policy scenarios developed by NREL to represent a range of potential future emissions factors, including (1) a "Current Policy" scenario, which uses median assumptions for model inputs, assumes that no new policies incentivizing decarbonization are passed, and includes the impacts of the Inflation Reduction Act passed in August of 2022, (2) a counterfactual scenario in which the Inflation Reduction Act was not passed, (3) and a scenario that includes the extension of two key tax credits for renewable energy generation that are currently scheduled for phase-out beginning in 2032. For consistency with the 2018 estimate based on eGRID data, we scaled state-level 2020 eGRID estimates by the changes projected in the NREL scenarios. These emissions factors were used to project future emissions from irrigation energy use under each scenario, assuming no changes in energy demand or fuel reliance.

In addition, we developed a "pump electrification" projection, in which we assumed an annual 5% decline in usage of each non-electric fuel in terms of energy content in each state. We assumed that the reduction in non-electric energy would be offset by a commensurate increase in electricity use, after controlling for standard estimates of fuel-specific motor efficiencies[44] (25% for propane, 23% for gasoline, 21% for natural gas, 31% for diesel, and 88% for electricity). We

combined these projections of pump fuel-switching with each of the NREL policy scenarios to explore the potential impacts of pump electrification.

## Data availability

All source data associated with this manuscript are publicly available as described in Supplementary Table S2. All data generated as part of this study are publicly available in Supplementary Table 3 and via Zenodo at https://doi.org/10.5281/zenodo.10416689.

## Code availability

All code associated with this manuscript is publicly available via Zenodo at https://doi.org/10.5281/zenodo.10416689.

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

## Acknowledgements

This study was supported by the National Science Foundation (DGE-1828902 and DGE-006784 to AWD and CBET-2144169 to L.T.M.), the Foundation for Food and Agriculture Research (FF-NIA19-0000000003 to NDM and FF-NIA19-0000000084 to L.T.M.), the United States Department of Agriculture National Institute for Food and Agriculture (2021-68014-34141 to N.D.M. and E.C.). Any opinions, findings, and conclusions or recommendations expressed in this material are those of the author(s) and do not necessarily reflect the views of the National Science Foundation or the Foundation for Food and Agriculture Research.

## Author contributions

N.D.M., R.T.C., A.W.D., and L.T.M. conceived and designed the study, and A.W.D. and E.C. conducted the data analyses. A.W.D. wrote the initial draft and all authors contributed to the final manuscript.

## Competing interests

The authors declare no competing interests.
