## [Peer Review File · Nature Communications]

Greenhouse gas emissions from US irrigation pumping and implications for climate-smart irrigation policyREVIEWER COMMENTS

Reviewer #1 (Remarks to the Author):

General: This paper builds directly on work by Sowby and Dicataldo (reference 17), which should be more thoroughly acknowledged. They were the first to 1) combine USDA's surveys and EIA's price data to derive irrigation-related energy use and 2) provide a national estimate of this energy-for-water activity, both of which have enabled the authors' study. The authors have filled several of the gaps Sowby and Dicataldo explicitly highlighted, including emissions, spatial sensitivity, and climate policy. As such, the authors' work is an excellent contribution.

Abstract: The abstract is well written and focused. It concisely presents the problem, scope, results, and implications.

Line 33: Other relevant energy-for-water studies are:

- Geographic Footprint of Electricity Use for Water Services in the Western U.S., <https://doi.org/10.1021/es5016845>. This one deals with irrigation, among other uses.
- The State of U.S. Urban Water: Data and the Energy-Water Nexus, <https://doi.org/10.1002/2017WR022265>
- Survey of Energy Requirements for Public Water Supply in the United States, <https://doi.org/10.5942/jawwa.2017.109.0080>
- Evaluating the Energy Intensity of the US Public Water System, <https://doi.org/10.1115/ES2011-54165>

•
Methods: Why is this section at the end? Unless this is a journal requirement, it should precede Results.

Methods: It is important to explain that irrigators don't report their energy use to anyone, so it has to be inferred by other means, namely, energy bills.

Results: In addition to the total emissions of 12.4 million metric tons CO₂e, the authors should report the total energy. Sowby and Dicataldo estimate 60.6 TWh of energy for irrigation in 2018 (44.0 TWh on-farm only), and this would be a worthwhile comparison. Using national average eGRID data, their emissions would come to 19.0 million metric tons (on-farm only), which is on the same order of magnitude as the authors' result (but without the spatial sensitivity).

Lines 137–138 and Figure S4: Where did pump efficiency information come from? USDA?

References: The references come from a variety of credible scientific materials.

All map figures: State outlines may improve readability, especially in the supplemental figures with so many blank spots.

Reviewer #2 (Remarks to the Author):

This manuscript fills an important gap: clarifying just how much CO₂ is being emitted by agricultural irrigation in the US. They take a "top down" approach, using survey data sources for energy and water use, along with water source and pump technology, to quantify first state-level, the county-level CO₂e emissions. Further, they examine crop-level CO₂ emissions, and look at how those vary across the US. Finally, they provide a time-varying set of scenarios out to 2050, examining the effects of recent policy changes (the IRA in the US), and potential electrification of pump technology (at a rate of 5% adoption per year).

I believe that this manuscript makes an important contribution to the literature, and is of sufficient merit and potential impact to be published in Nature Communications. However, there are three issues that I believe should be addressed prior to publication. These may be addressed either in more thorough Discussion or in adding estimates of missing CO₂ emission sources.

First, I'd like to identify myself. I am Anthony Kendall, a hydrogeologist at Michigan State University. In 2020, I co-authored a paper led by Ben McCarthy looking at energy and CO₂ emissions from irrigation in the Kansas High Plains Aquifer using a "bottom up" approach: computing the energy requirements and emissions at each point of diversion over the aquifer in that state. Thank you for your work, I hope it is found suitable for publication following revision.

Primary Issues

The paper reads very well, balancing the presentation of a large quantity of numbers (and 90% confidence intervals) with compelling text and tight topic sentences. The figures are well constructed, and I found the Supplement to be plenty informative.

As alluded to above, I have three primary issues. The first is that the authors do not compare their findings to those of any other work, regional or nationwide. My second issue is that, even though they bring up other sources of CO₂ emissions in irrigated agriculture other than those from the pump energy source, they do not detail or discuss the magnitude of those. Third, the authors do not discuss the limitations of their work, nor suggest future work to improve upon their own.

Lack of Comparison to Other Sources

The authors have identified what they (and I) believe to be a first-of-its kind analysis of the CO₂ emissions from irrigated agriculture across the US. In such an instance, it can be difficult to provide comparisons to other studies, given the mismatch in scale or methods employed. However, the authors do not attempt to do so in this case.

There are several options:

- 1) Compare results to similar studies conducted elsewhere. Without a literature review, I don't know if CO₂ emissions have been cataloged in irrigation in the EU, China, India, or elsewhere. Yet I suspect that is the case, and meaningful comparison could be made. This could also provide further context for the numbers (including the regional variability) presented here. This sort of comparison would also serve to broaden the audience for this work. Nature Communications is a global journal, and although US CO₂ emissions are an obviously important part of the overall global budget, they are just one region. A relatively small effort could bring in a broader global audience, and acknowledge the work of others in this field.
- 2) Compare a portion of their study domain to the results of regional studies. At the very least, the authors cite McCarthy et al. 2020, which produced a CO₂e/m³ pumped estimate which can be compared to their own. They report a nationwide average of 152g CO₂e/m³ (line 85). For comparison, we found 250 - 270g CO₂e/m³ for the Kansas HPA. The authors can compare their Kansas values more specifically to show that the top-down and bottom-up approaches yield similar results. There may be other such regional studies as well.
- 3) Compare an aspect of their work (energy use) with the estimates of others. If I am understanding correctly, in Equation 1 the quotient $D \cdot A / P$ is energy use. Thus, this study implicitly produces energy use estimates which could then be compared to those (citations 15 and 16) of others. I am not certain if this would simply be a duplication of the methods of those other papers, but some effort to compare should be made.

No Consideration of other CO₂ sources

I got excited when, in the second paragraph (lines 43 and 44), the authors mentioned three additional CO₂ sources from irrigation in a fashion that suggested they would be taking these into account. But, alas that was not done. The three sources they mention are: 1) increased N₂O emissions from wetter

soils, 2) CH₄ emissions from surface reservoirs, and 3) CO₂ emissions from groundwater supersaturated in bicarbonate.

I haven't researched the magnitude of the first two, but from our work in the Kansas HPA I know that the third factor accounts for approximately 30% of overall CO₂ emissions. Anywhere that groundwater is being depleted by pumping (which is, in reality, most everywhere), groundwater has to equilibrate with the atmosphere. In this process, as pointed out by Wood and Hyndman (2017), a lot of CO₂ is released. The exact amount varies according to groundwater bicarbonate concentrations. But the methods exist to calculate this factor. It would likely exacerbate the extreme regional imbalances of CO₂ emissions in areas with deep, declining aquifers.

I understand that every study has its limits, and that's fine. But, you have to acknowledge those limits, and discuss the implications of them (see next section).

No Limitations or Future Work

There should be some discussion throughout their results/discussion section of the limitations of their work. For instance:

- * The work relies entirely on survey data that can be incomplete or biased. The Ag Census, for instance, has had significant declines in its response rate. The 2018 Irrigation and Water Management survey had a response rate of 64.4%, a decline from the 69.8% rate from the 2013 Farm and Ranch Irrigation Survey (the IWM's predecessor). While the USDA makes every effort to correct for this, there are big financial incentives for farmers to report certain numbers, which may deviate from actual use.
- * The mismatch between the USGS and USDA water use numbers is substantial--as they discuss in their Supplement, and note in the main document on Lines 159 and 160. The result of this is that the absolute values are likely low, but regional variation is probably consistent. This (or other) implications of their limitations should be addressed more directly.
- * Lack of consideration of other emissions factors (see my previous issue) likely means that there are regional biases in the underestimation of CO₂e emissions from irrigation.

Suggestions for future work follow from limitations, along with the scope or geographic boundaries drawn in the work. These suggestions are invaluable to help others identify gaps in the science--pointing the way for the next generation of researchers to build on your work. Please add some. You are more knowledgeable than almost everyone else in this field right now, share your deep expertise with other researchers!

Reviewer #3 (Remarks to the Author):

This article makes an assessment of the potential conflicts between adaptive irrigation expansion and agricultural emissions mitigation. The emission estimates are calculated at the state level and then apportioned at the county level, by energy source, water source and crop. The policy implications are drawn with respect to "climate-smart" initiatives. A spatially distributed estimation of sectoral (irrigation) emissions would indeed be informative for policy and research. I look forward to a version of this work being published in the near future.

I understand the format of the journal may call to organize the article such that results are presented immediately following the introduction. However, given that the results are the product of a non-standard procedure, it would benefit general readability to discuss the methods before providing the results. It was hard for me to stay focused and follow the results discussion with the burning question of how they've arrived at those numbers to start with.

The core of the exercise is to estimate emissions at the state level (Eq.1). The approach is straight forward and, in my opinion, valid. However, the estimates depend critically on accurate data on per acre expenditures on energy use for irrigation (farmer survey-based), price of energy by type

(EIA/PADD) and irrigated acreage (USDA). There is not much that can be done with respect to survey data as reported which is as best one can probably do with available information. However, the distortion introduced by the reference prices can have a significant impact on both the overall level and allocation by energy type of emissions. Specifically, the average annual price for No.2 Diesel is likely underestimating the level of emissions from Diesel because irrigation pumps typically employ off-road diesel which is significantly cheaper than on-road diesel, which equates to more water use and emissions from this energy type. This price can be corrected by subtracting the fuel taxes by state from the existing price levels. At least in the area that I am familiar with, irrigation electricity rates are higher than industrial rates (huge variation across the country, though) which would over estimate the amount of water use and emissions from this energy source—unfortunately I do not have a quick and easy way to adjust and it may not be the case across the country. A similar issue may arise with the other energy sources. Another important implication with respect to the results that derive from this would be the amount of emissions from ground versus surface water.

The state emissions estimates are then downscaled proportionally by source of water (Source/Total) with a further relative adjustment for groundwater based on average pumping depth (County/ State). I believe this is a valid inter-county allocation approach. One robustness check that I would suggest is to calculate the amount of energy required to pump groundwater for a given depth using a benchmark such as the Nebraska Pumping Standard and compare the resulting energy use by type against the given estimates.

Because the article attempts to provide a spatial landscape of emissions, it would help to add county and state demarcation to the maps. Furthermore, the narrative must include a discussion of the Colorado River Basin in the accounting of groundwater vs surface water use and emissions. The proportion within the basin would be 3 (surface) to 1 (groundwater) within the basin versus majority groundwater elsewhere. Consequently, the national median may not be the best benchmark to compare results against and adding Colorado basin and non-CB could be informative.

Perhaps the most interesting thought experiment in the article is the emissions projection based on electrification trends. In this regard, a one-year estimate seems insufficient to make projections. It seems appropriate to have at least two data points to project a line. Furthermore, given the identified correlation between electric pumps and groundwater use, I wonder if the increased in pumping lifts from depleting aquifers is considered for these projections. These details must be made transparent in article to provide adequate caveats to the estimates.

Doing a quick search on the topic and taking a couple of first-page articles, it appears that more discussion on the current state of knowledge is required. Rothausen and Conway (2011) provide estimates of irrigation sector energy use or emissions from various countries which would be worthwhile over-viewing for benchmark to your results. Sapkota et al. (2020) provide micro-estimates of irrigation GHG emissions (field-level) which should be used to benchmark the deduced macro-estimates in your article. I'm no longer convinced that being the only ones providing estimates is sufficient for publication without discussion as to why these estimates haven't been presented before (difficulties, biases, etc.) or a good overview of the current state of knowledge on the topic, from whichever region.

- Rothausen, S.G. and Conway, D., 2011. Greenhouse-gas emissions from energy use in the water sector. *Nature Climate Change*, 1(4), pp.210-219.

- Sapkota, A., Haghverdi, A., Avila, C.C. and Ying, S.C., 2020. Irrigation and greenhouse gas emissions: a review of field-based studies. *Soil Systems*, 4(2), p.20.

There must be others.

REVIEWER COMMENTS

Reviewer #1 (Remarks to the Author):

General: This paper builds directly on work by Sowby and Dicaldo (reference 17), which should be more thoroughly acknowledged. They were the first to 1) combine USDA's surveys and EIA's price data to derive irrigation-related energy use and 2) provide a national estimate of this energy-for-water activity, both of which have enabled the authors' study. The authors have filled several of the gaps Sowby and Dicaldo explicitly highlighted, including emissions, spatial sensitivity, and climate policy. As such, the authors' work is an excellent contribution.

We thank the reviewer for their supportive comments, and we very much appreciate the pioneering work of Sowby and Dicaldo in this space. We have added and/or revised the following sections throughout the manuscript to 1) highlight their contribution more explicitly and 2) more clearly distinguish this work from previous energy-for-water studies. We have also added a table containing estimates of irrigation energy use and emissions from selected previous studies from the US (including from Sowby & Dicaldo 2022) and internationally to provide additional context for our findings.

Lines 60-70: While there have been previous efforts to quantify energy use in the US water sector, they have often focused on the public water supply rather than irrigation^{17,24-26} despite the fact that water withdrawals for irrigation are approximately three times larger than those for the public supply²⁷. Several studies have explicitly evaluated energy use for irrigation pumping, but these studies have generally focused on regional assessments²⁸, exclusively considered electrical use²⁹, are not spatially resolved³⁰, and/or have not estimated associated GHG emissions. Here, we provide the first national-scale, spatially explicit estimate of GHG emissions from energy use for irrigation and assess implications of recent climate policy for reducing these emissions.

Lines 93-107: This corresponds to an estimated 153 PJ of total energy use for on-farm irrigation pumping. Previous studies have estimated on-farm irrigation pump energy use at 158 PJ nationally³⁰ and 136 PJ for electricity use in the Western USA²⁹, in close agreement with our estimates. Table S1 provides energy and emissions intensity estimates from selected regional and international studies, demonstrating substantial variability in estimated intensities. On a per-hectare basis, for instance, energy intensity estimates have ranged from 6,542 MJ ha⁻¹ (our study) up to 43,412 MJ ha⁻¹ (from a study of groundwater irrigation in Pakistan), while emissions intensity estimates have varied from 0.53 tonnes CO₂e ha⁻¹ (our study) up to 1.27 tonnes CO₂e ha⁻¹ (from a study of groundwater irrigation in India). Previous studies have often focused on groundwater pumping, which is associated with higher energy use and subsequently higher emissions. Additionally, most of these studies rely on a bottom-up approach, using information about pumping depths, volumes, and efficiencies to calculate the theoretical pump energy requirements. Further comparisons between bottom-up and top-down approaches (such as that

used in this manuscript) would be useful to increase confidence in irrigation energy use estimates.

Lines 434-436: This approach broadly follows that of Sowby and Dicataldo²⁸, who first presented national-scale estimates of irrigation energy use by integrating data on pump fuel expenditures with fuel prices.

Abstract: The abstract is well written and focused. It concisely presents the problem, scope, results, and implications.

Line 33: Other relevant energy-for-water studies are:

- **Geographic Footprint of Electricity Use for Water Services in the Western U.S.**, <https://doi.org/10.1021/es5016845>. This one deals with irrigation, among other uses.
- **The State of U.S. Urban Water: Data and the Energy-Water Nexus**, <https://doi.org/10.1002/2017WR022265>
- **Survey of Energy Requirements for Public Water Supply in the United States**, <https://doi.org/10.5942/jawwa.2017.109.0080>
- **Evaluating the Energy Intensity of the US Public Water System**, <https://doi.org/10.1115/ES2011-54165>

Thank you for these suggestions. We have added these citations to the introduction, where we expanded discussion of previous work on the energy-water nexus (lines 60-70, see response to first comment).

Methods: Why is this section at the end? Unless this is a journal requirement, it should precede Results.

We have placed the methods at the end per the formatting requirements listed in the Nature Communications Guide to Authors (<https://www.nature.com/ncomms/submit/article>).

Methods: It is important to explain that irrigators don't report their energy use to anyone, so it has to be inferred by other means, namely, energy bills.

We have clarified this in the methods section as follows:

Lines 450-451: Because irrigators do not directly report energy use, we rely on integration of expenditures and fuel prices as a proxy for energy use.

Results: In addition to the total emissions of 12.4 million metric tons CO₂e, the authors should report the total energy. Sowby and Dicataldo estimate 60.6 TWh of energy for irrigation in 2018 (44.0 TWh on-farm only), and this would be a worthwhile comparison. Using national average eGRID data, their emissions would come to 19.0 million metric tons (on-farm

only), which is on the same order of magnitude as the authors' result (but without the spatial sensitivity).

Thank you for suggesting this comparison of total energy use. In total, we estimate 42.5 TWh of energy use, which is remarkably similar to the Sowby & Dicaldo on-farm energy use estimate of 44.0 TWh annually. We have added the following text to report energy use and compare with previous estimates:

Lines 93-96: This corresponds to an estimated 153 PJ of total energy use for on-farm irrigation pumping. Previous studies have estimated on-farm irrigation pump energy use at 158 PJ nationally³⁰ and 136 PJ for electricity use in the Western USA²⁹, in close agreement with our estimates.

Lines 137–138 and Figure S4: Where did pump efficiency information come from? USDA?

Average motor fuel efficiency values (25% for propane, 23% for gasoline, 21% for natural gas, 31% for diesel, and 88% for electricity) were taken from the following University of Nebraska conference publication, as noted in lines 582-583.

Martin, D. L., Dorn, T. W., Melvin, S. R., Corr, A. J. & Kranz, W. L. Evaluating energy use for pumping irrigation water. Proceedings of the 23rd Annual Central Plains Irrigation Conference (2011).

Although operational pump efficiency is tied to a variety of factors such as maintenance, weather, operating pressure and flow, etc., we believe that the average efficiency factors presented by Martin et al. provide a useful baseline for checking for an association between fuel selection and emissions. We have clarified the relevant text and added an additional reference to the data source in the results:

Lines 192-193: Average pump fuel efficiency, which is much higher for electrical pumps (88%) than propane, gasoline, natural gas, or diesel pumps (25%, 23%, 21%, and 31%, respectively)⁴⁵, was also strongly negatively associated with emissions (Figure 2D, Figure S3B, Figure S4).

References: The references come from a variety of credible scientific materials.

All map figures: State outlines may improve readability, especially in the supplemental figures with so many blank spots.

We agree and have revised all of the maps to include state outlines.

Reviewer #2 (Remarks to the Author):

This manuscript fills an important gap: clarifying just how much CO₂ is being emitted by agricultural irrigation in the US. They take a "top down" approach, using survey data sources

for energy and water use, along with water source and pump technology, to quantify first state-level, the county-level CO₂e emissions. Further, they examine crop-level CO₂ emissions, and look at how those vary across the US. Finally, they provide a time-varying set of scenarios out to 2050, examining the effects of recent policy changes (the IRA in the US), and potential electrification of pump technology (at a rate of 5% adoption per year).

I believe that this manuscript makes an important contribution to the literature, and is of sufficient merit and potential impact to be published in Nature Communications. However, there are three issues that I believe should be addressed prior to publication. These may be addressed either in more thorough Discussion or in adding estimates of missing CO₂ emission sources.

First, I'd like to identify myself. I am Anthony Kendall, a hydrogeologist at Michigan State University. In 2020, I co-authored a paper led by Ben McCarthy looking at energy and CO₂ emissions from irrigation in the Kansas High Plains Aquifer using a "bottom up" approach: computing the energy requirements and emissions at each point of diversion over the aquifer in that state. Thank you for your work, I hope it is found suitable for publication following revision.

Dr. Kendall, thank you very much for your constructive feedback, which we believe has substantially improved this manuscript. Your work on irrigation dynamics, climate interactions, and irrigation emissions has been highly influential to the development of this project, and we appreciate your work in this area!

Primary Issues

The paper reads very well, balancing the presentation of a large quantity of numbers (and 90% confidence intervals) with compelling text and tight topic sentences. The figures are well constructed, and I found the Supplement to be plenty informative.

As alluded to above, I have three primary issues. The first is that the authors do not compare their findings to those of any other work, regional or nationwide. My second issue is that, even though they bring up other sources of CO₂ emissions in irrigated agriculture other than those from the pump energy source, they do not detail or discuss the magnitude of those. Third, the authors do not discuss the limitations of their work, nor suggest future work to improve upon their own.

Lack of Comparison to Other Sources

The authors have identified what they (and I) believe to be a first-of-its kind analysis of the CO₂ emissions from irrigated agriculture across the US. In such an instance, it can be difficult to provide comparisons to other studies, given the mismatch in scale or methods employed. However, the authors do not attempt to do so in this case.

There are several options:

1) Compare results to similar studies conducted elsewhere. Without a literature review, I

don't know if CO₂ emissions have been cataloged in irrigation in the EU, China, India, or elsewhere. Yet I suspect that is the case, and meaningful comparison could be made. This could also provide further context for the numbers (including the regional variability) presented here. This sort of comparison would also serve to broaden the audience for this work. Nature Communications is a global journal, and although US CO₂ emissions are an obviously important part of the overall global budget, they are just one region. A relatively small effort could bring in a broader global audience, and acknowledge the work of others in this field.

2) Compare a portion of their study domain to the results of regional studies. At the very least, the authors cite McCarthy et al. 2020, which produced a CO₂e/m³ pumped estimate which can be compared to their own. They report a nationwide average of 152g CO₂e/m³ (line 85). For comparison, we found 250 - 270g CO₂e/m³ for the Kansas HPA. The authors can compare their Kansas values more specifically to show that the top-down and bottom-up approaches yield similar results. There may be other such regional studies as well.

3) Compare an aspect of their work (energy use) with the estimates of others. If I am understanding correctly, in Equation 1 the quotient $D \cdot A / P$ is energy use. Thus, this study implicitly produces energy use estimates which could then be compared to those (citations 15 and 16) of others. I am not certain if this would simply be a duplication of the methods of those other papers, but some effort to compare should be made.

This is an excellent suggestion, and one that was echoed by both other reviewers as well. In hopes of addressing this comprehensively, we have compiled a table of published estimates of total energy use, total GHG emissions, and per-hectare and per-volume emissions and energy intensities for irrigation pumping, as available. As suggested, we have done this by leveraging global and regional studies, as well as those that exclusively produced energy use estimates. We have included this information as a supplementary table (Table S1) and have also added a paragraph of text situating our results within the context of previous estimates:

Lines 93-107: This corresponds to an estimated 156 PJ of total energy use for on-farm irrigation pumping. Previous studies have estimated on-farm irrigation pump energy use at 158 PJ nationally³⁰ and 136 PJ for electricity use in the Western USA²⁹, in close agreement with our estimates. Table S1 provides energy and emissions intensity estimates from selected regional and international studies, demonstrating substantial variability in estimated intensities. On a per-hectare basis, for instance, energy intensity estimates have ranged from 6,687 MJ ha⁻¹ (our study) up to 43,412 MJ ha⁻¹ (from a study of groundwater irrigation in Pakistan), while emissions intensity estimates have varied from 0.54 tonnes CO₂e ha⁻¹ (our study) up to 1.27 tonnes CO₂e ha⁻¹ (from a study of groundwater irrigation in India). Previous studies have often focused on groundwater pumping, which is associated with higher energy use and subsequently higher emissions. Additionally, most of these studies rely on a bottom-up approach, using information about pumping depths, volumes, and efficiencies to calculate the theoretical pump energy requirements. Further comparisons between bottom-up and top-down approaches (such as that used in this manuscript) would be useful to increase confidence in irrigation energy use estimates. Additionally, most of these studies rely on a bottom-up approach, using information about pumping depths, volumes, and efficiencies to calculate the theoretical pump energy

requirements. Further comparisons between bottom-up and top-down approaches (such as that used in this manuscript) would be useful to increase confidence in irrigation energy use estimates.

No Consideration of other CO₂ sources

I got excited when, in the second paragraph (lines 43 and 44), the authors mentioned three additional CO₂ sources from irrigation in a fashion that suggested they would be taking these into account. But, alas that was not done. The three sources they mention are: 1) increased N₂O emissions from wetter soils, 2) CH₄ emissions from surface reservoirs, and 3) CO₂ emissions from groundwater supersaturated in bicarbonate.

I haven't researched the magnitude of the first two, but from our work in the Kansas HPA I know that the third factor accounts for approximately 30% of overall CO₂ emissions. Anywhere that groundwater is being depleted by pumping (which is, in reality, most everywhere), groundwater has to equilibrate with the atmosphere. In this process, as pointed out by Wood and Hyndman (2017), a lot of CO₂ is released. The exact amount varies according to groundwater bicarbonate concentrations. But the methods exist to calculate this factor. It would likely exacerbate the extreme regional imbalances of CO₂ emissions in areas with deep, declining aquifers.

I understand that every study has its limits, and that's fine. But, you have to acknowledge those limits, and discuss the implications of them (see next section).

We agree that consideration of these additional sources is very important for developing a comprehensive picture of irrigation-related emissions. For the following reasons, we have decided that these emissions are beyond the scope of the present analysis:

1. *First and foremost, there is substantial methodological complexity and uncertainty associated with estimating emissions from these alternative sources. We felt that deploying and thoroughly explaining four very diverse methods for source-specific emissions estimations would have resulted in excessive complexity for a single manuscript. Although there is also uncertainty in the survey data utilized for the present study, we feel that it is well-constrained and relatively straightforward to estimate given that coefficients of variation are provided by the USDA for the state-level fuel expenditure data. Specifically:*
 - a. *For N₂O, Tier 1 emissions factors have been estimated in only a small handful of irrigated systems in the US (for example, three locations were included in the comprehensive database compiled by Hergoualc'h et al. in 2021). Ecosystem biogeochemical models, as are used for Tier 3 estimates in the US GHG Inventory, are better equipped to represent the high spatial and temporal heterogeneity of N₂O emissions. However, deployment of these models at the national scale requires an enormous amount of input data, parameter tuning, and expertise. Additionally, structural and input data uncertainty in the DayCent biogeochemical model is large nationally, and*

even larger at smaller spatial scales. Nationally, the 95% confidence interval for total N₂O emissions (133-304 Gg N₂O-N) represents 85% of the magnitude of the central estimate (201 Gg N₂O-N) (del Grosso et al 2010). This uncertainty would inevitably propagate to estimates of the specific contribution of irrigation.

- b. Similarly, as you are well aware, estimation of CO₂ degassing from supersaturated aquifers requires interpolation of measured bicarbonate concentrations or calculation of groundwater pCO₂ from alternative measurements. While some regions have dense monitoring networks, measurements of both groundwater chemistry and depletion are sparse in much of the nation. Interpolation of groundwater bicarbonate concentrations and depletion, particularly where measurements are sparse, may lead to potentially large but unquantifiable uncertainties. Moreover, these calculations are subject to the assumptions outlined in Wood & Hyndman (2017), specifically that 50% of the additional carbon originates from microbially-derived CO₂ in soil air and 50% from interactions with the carbonate matrix. However, these assumptions are based on generalized reactions rather than explicit tracing of C origin and fate.
 - c. Finally, attribution of reservoir methane emissions explicitly to irrigation is difficult, given that most reservoirs serve a myriad of roles such as public or industrial supply, hydroelectric generation, recreation, habitat, and flood control, among others. To the best of our knowledge, there is no national-scale effort to track end uses of water stored in reservoirs.
2. A primary focus of the present manuscript is to evaluate decarbonization potential of energy use for irrigation, including impacts of the Inflation Reduction Act. Unfortunately, decarbonization options for non-fossil fuel emissions sources are much more limited. Appropriate N management and minimizing time under saturated conditions (70-90% water filled pore space) may help mitigate N₂O emissions, while reduced groundwater extraction and depletion would help mitigate degassing emissions. However, policy levers for implementing these changes are limited.

That said, to better acknowledge these sources and their potential magnitude, we have added a paragraph focused on limitations and future directions, which includes following text:

Lines 377-406: Importantly, reducing emissions from irrigation will require more than simply decarbonization of energy use for on-farm pumping. Several additional emissions sources contribute to the total GHG footprint of irrigation, such as elevated N₂O emissions due to increased soil moisture, degassing of groundwater supersaturated in CO₂ and N₂O, energy use for off-farm pumping, and CH₄ production from reservoirs used for irrigation. Thus, the total GHG emissions impact of irrigation is higher than the values presented here, which include only emissions from on-farm pumping. CO₂ degassing from groundwater depletion for all end uses has been estimated at 1.7 MMT CO₂e annually²¹, while field-scale studies have identified

increases in N₂O emissions of up to 170% under irrigated vs. rainfed conditions⁴⁶. We expect spatial patterns of these additional emissions to vary, exacerbating total irrigation-related emissions in areas with extensive groundwater depletion (due to groundwater CO₂ degassing), in arid areas (due to larger impacts on soil moisture and subsequently N₂O emissions), and in areas with heavy reliance on imported water (due to pumping requirements and extensive reservoir storage). Mitigation of emissions from these alternative sources is likely to be more complex than mitigation of energy use emissions; a comprehensive quantification will help guide GHG reduction efforts.

Finally, we are currently working on a follow-up manuscript that addresses emissions from additional sources, including a) N₂O emissions using point-level estimates from the US GHG Inventory, b) CO₂ emissions from energy use for inter-basin transfers using data collected directly from transfer operators, and c) CO₂ emissions from groundwater degassing using a USGS dataset of well water alkalinity, pH, and salinity coupled with data on groundwater extraction and depletion. Despite the uncertainties, we hope that this follow-up manuscript will provide an initial comprehensive estimate of irrigation-related emissions in the US and serve as a starting point for deeper analysis of these alternative sources. Preliminary results from this study suggest that energy use for on-farm pumping is the dominant emissions source from irrigation, representing roughly two thirds of total emissions.

Works referenced:

Hergoualc'h, K. *et al.* Improved accuracy and reduced uncertainty in greenhouse gas inventories by refining the IPCC emission factor for direct N₂O emissions from nitrogen inputs to managed soils. *Global Change Biology* **27**, 6536–6550 (2021).

Del Grosso, S. J., Ogle, S. M., Parton, W. J. & Breidt, F. J. Estimating uncertainty in N₂O emissions from U.S. cropland soils. *Global Biogeochemical Cycles* **24**, (2010).

Wood, W. W. & Hyndman, D. W. Groundwater Depletion: A Significant Unreported Source of Atmospheric Carbon Dioxide. *Earth's Future* **5**, 1133–1135 (2017).

No Limitations or Future Work

There should be some discussion throughout their results/discussion section of the limitations of their work. For instance:

We appreciate this feedback and have added text throughout the manuscript to discuss limitations and recommendations for future work, as detailed below.

*** The work relies entirely on survey data that can be incomplete or biased. The Ag Census, for instance, has had significant declines in its response rate. The 2018 Irrigation and Water Management survey had a response rate of 64.4%, a decline from the 69.8% rate from the 2013 Farm and Ranch Irrigation Survey (the IWM's predecessor). While the USDA makes every**

effort to correct for this, there are big financial incentives for farmers to report certain numbers, which may deviate from actual use.

Reliance on survey data is certainly a limitation to the analysis, particularly given relatively low response rates. We are very appreciative that the IWMS reports coefficients of variation alongside estimates of irrigation expenditures and acreage; these reported uncertainty values were included in the Monte Carlo bootstrap and thus are reflected in the confidence intervals reported.

Lines 444-450: Although survey data are subject to limitations, such as incomplete response rates and potential misreporting by respondents, they are the best available data at the national scale. Additionally, the IWMS provides coefficients of variation alongside the data, which were incorporated into the uncertainty analysis.

*** The mismatch between the USGS and USDA water use numbers is substantial--as they discuss in their Supplement, and note in the main document on Lines 159 and 160. The result of this is that the absolute values are likely low, but regional variation is probably consistent. This (or other) implications of their limitations should be addressed more directly.**

We have expanded on this limitation and its implications as follows to clarify that this discrepancy only affects the absolute magnitude of crop-specific emissions estimates:

Lines 225-231: This analysis required integration of water use data from both the USDA and the USGS, which have discrepancies in methodology and subsequent total estimates of irrigation water use, likely resulting in absolute estimates of crop-specific water use that are too low. However, the relative contribution of each crop, the spatial distribution of crop-specific estimates, and all non-crop-specific results are unaffected by the discrepancies between the USDA and USGS data. Further implications are detailed in the Supplementary Discussion.

*** Lack of consideration of other emissions factors (see my previous issue) likely means that there are regional biases in the underestimation of CO₂e emissions from irrigation.**

Inclusion of additional sources is indeed likely to alter the distribution of GHG emissions to some degree. We suspect that spatial patterns in many of these alternative sources will largely align with spatial patterns in irrigation pumping emissions. For instance, areas with high groundwater reliance have high energy requirements but also are likely to have high degassing emissions. Similarly, high N₂O and energy use emissions are expected to coincide in arid regions, where crop water demand is high and impacts on soil microbial metabolism tend to be larger. We hope that the addition of text discussing other sources that was added in response to the previous comment (lines 377-406), clarifies that the spatial patterns presented reflect only energy use emissions, not total irrigation emissions.

Suggestions for future work follow from limitations, along with the scope or geographic boundaries drawn in the work. These suggestions are invaluable to help others identify gaps

in the science--pointing the way for the next generation of researchers to build on your work. Please add some. You are more knowledgeable than almost everyone else in this field right now, share your deep expertise with other researchers!

Thank you for this recommendation! We have added the following text prior at the end of the results and discussion section to highlight some future directions of interest that arose from this work:

Lines 364-375: Further work is needed to understand and reduce infrastructural and economic barriers to pump electrification. Electrification requires an up-front investment in a new pumping system, a reliable connection to the electrical grid, and potentially higher operational costs depending on local energy prices. Strategic incentives for electric pump adoption could offset cost-related barriers and help to accelerate pump electrification. For areas where grid connectivity is limiting, expansion of solar pump usage may be needed to fully eliminate pumping emissions. Additionally, scenario analyses that integrate the effects of pump electrification and grid decarbonization with variables that will influence pump energy demand would be useful to clarify future projections of irrigation-related emissions. For instance, declining groundwater levels, migration of irrigation and crop types, changes to surface versus groundwater reliance, and changes to irrigation water demand due to climate change will all affect total pump energy requirements.

Reviewer #3 (Remarks to the Author):

This article makes an assessment of the potential conflicts between adaptive irrigation expansion and agricultural emissions mitigation. The emission estimates are calculated at the state level and then apportioned at the county level, by energy source, water source and crop. The policy implications are drawn with respect to “climate-smart” initiatives. A spatially distributed estimation of sectoral (irrigation) emissions would indeed be informative for policy and research. I look forward to a version of this work being published in the near future.

I understand the format of the journal may call to organize the article such that results are presented immediately following the introduction. However, given that the results are the product of a non-standard procedure, it would benefit general readability to discuss the methods before providing the results. It was hard for me to stay focused and follow the results discussion with the burning question of how they’ve arrived at those numbers to start with.

We have included the methods at the end of the manuscript to comply with the journal’s author guidelines (<https://www.nature.com/ncomms/submit/guide-to-authors>). In order to orient the reader to our approach, we do provide a summary of the methods at the end of the introduction (lines 74-84). For now, we will defer to the editor’s judgement regarding if the length and level of detail in this methods summary is appropriate.

The core of the exercise is to estimate emissions at the state level (Eq.1). The approach is straight forward and, in my opinion, valid. However, the estimates depend critically on accurate data on per acre expenditures on energy use for irrigation (farmer survey-based), price of energy by type (EIA/PADD) and irrigated acreage (USDA). There is not much that can be done with respect to survey data as reported which is as best one can probably do with available information. However, the distortion introduced by the reference prices can have a significant impact on both the overall level and allocation by energy type of emissions. Specifically, the average annual price for No.2 Diesel is likely underestimating the level of emissions from Diesel because irrigation pumps typically employ off-road diesel which is significantly cheaper than on-road diesel, which equates to more water use and emissions from this energy type. This price can be corrected by subtracting the fuel taxes by state from the existing price levels. At least in the area that I am familiar with, irrigation electricity rates are higher than industrial rates (huge variation across the country, though) which would over estimate the amount of water use and emissions from this energy source—unfortunately I do not have a quick and easy way to adjust and it may not be the case across the country. A similar issue may arise with the other energy sources. Another important implication with respect to the results that derive from this would be the amount of emissions from ground versus surface water.

We have added the following text to acknowledge limitations associated with reliance on survey data:

Lines 444-451: Although survey data are subject to limitations, such as incomplete response rates and potential misreporting by respondents, they are the best available data at the national scale. Additionally, the IWMS provides coefficients of variation alongside the data, which were incorporated into the uncertainty analysis. Because irrigators do not directly report energy use, we rely on integration of energy expenditures and energy prices as a proxy for energy use.

We had not considered the impact of taxes on diesel estimates, and the EIA data do include federal and state excise taxes in diesel and gasoline price estimates. Thank you for pointing out this oversight. The EIA does not publish data specifically on off-road diesel or gasoline prices. Based on your comment, we have re-estimated diesel and gasoline prices by subtracting federal and state-level excise taxes from EIA on-road diesel prices. This recalculation caused a 0.16 MMT increase in our estimate of total annual emissions, and the share of emissions from diesel increased from 8.9% to 10.6%. We have added the following text to the methods to clarify this additional step, have added the tax dataset to Table S2, and have changed all affected values and figures related to emissions and energy use throughout the manuscript:

Lines 468-471: The EIA data for diesel and gasoline include federal and state excise taxes that are applicable to on-road users but not agricultural users. To better reflect off-road diesel and gasoline prices, 2015 federal and state taxes from the EIA were subtracted from the on-road prices.

We are aware of the variability in electricity rate setting across different utilities. The EIA “industrial” prices are intended to represent rates for agricultural users, based on our correspondence with EIA representatives, and we believe that these are the best available data for our calculations. However, we have also added the following text to acknowledge limitations associated with the electricity price data:

Lines 471-475: For electricity prices, we used EIA prices for industrial users. These are the best available data to our knowledge, but we acknowledge that there is substantial variation in electrical pricing between utilities. Thus, the prices paid by irrigators may diverge from the EIA estimates, and development of energy price estimates specific to agricultural users may improve the analysis.

The state emissions estimates are then downscaled proportionally by source of water (Source/Total) with a further relative adjustment for groundwater based on average pumping depth (County/ State). I believe this is a valid inter-county allocation approach. One robustness check that I would suggest is to calculate the amount of energy required to pump groundwater for a given depth using a benchmark such as the Nebraska Pumping Standard and compare the resulting energy use by type against the given estimates.

We agree that comparisons between bottom-up estimates, which calculate energy requirements based on pumping depth, pumping volumes, and pump efficiencies, and top-down estimates such as ours can be very informative, and we appreciate the encouragement to investigate such a comparison. In response, we estimated county-level pump energy use based on water table depth using the following bottom-up equation from Rothausen & Conway (2011, Nat. Clim. Change):

$$\text{Energy use (kWh)} = \frac{0.0027 \times h \times V}{\eta}$$

Where h is equal to pump head (in meters), V is equal to the volume of water pumped (in cubic meters), and η is equal to pump efficiency (%). Our data are unfortunately very limited in their capacity to accurately represent pump lift and efficiency. For lift, we rely on the modeled water table depth data. However, the actual pumping depth is likely to be deeper (potentially much deeper) than the water table depth. Friction losses would also need to be added to accurately account for total pump head. This suggests that our bottom-up estimates will be underestimated relative to our top-down estimates, as the calculated bottom-up estimates are representing shallower pumping depths with zero friction losses.

Similarly, our measure of “pump efficiency” used elsewhere, including Figures S1 and S4, accounts only for motor *fuel* efficiency (i.e., the proportion of potential energy in the fuel that is converted to mechanical energy by the motor). Overall pump efficiency is influenced by many, many additional factors, including variables like pump design, maintenance, weather, flow rates, and operating pressure. We lack the national data on overall pump efficiencies that would be needed to develop accurate bottom-up estimates. In responding to this comment, we realized

that our use of the term “pump efficiency” was potentially misleading and have changed it to “pump fuel efficiency” throughout the manuscript. In lieu of additional data, we use our pump fuel efficiency estimates for this bottom-up estimate of energy use. However, these efficiencies assume 100% efficiency in all other components of the pumping system, which is certainly not the case. Together, underestimation of pumping depths and overestimation of pump efficiency compound to produce bottom-up estimates that are lower than our top-down estimates. An example of this underestimation for California is provided below.

We regret that additional data are not available to develop more accurate bottom-up estimates at the national scale in this study. Because of the systematic underrepresentation inherent in the available data, we think that this bottom-up estimation does not provide a useful comparison to the top-down approach utilized throughout the manuscript. However, our approach is similar to the standard bottom-up approach in that we assume a linear relationship between pumping depth and energy use. We have added the following text highlighting that a thorough comparison between bottom-up and top-down approaches would be valuable:

Line 103-107: Additionally, most of these studies rely on a bottom-up approach, using information about pumping depths, volumes, and efficiencies to calculate the theoretical pump energy requirements. Further comparisons between bottom-up and top-down approaches (such as that used in this manuscript) would be useful to increase confidence in irrigation energy use estimates.

Finally, a previous study of the Kansas High Plains Aquifer took a bottom-up approach to provide a detailed calculation of energy use in the region based on best-available local data. We found strong agreement between these two separate methods, providing us with additional confidence that our groundwater depth scaling approach is reasonable. We have added the following text to highlight this comparison:

Lines 200-203: Our estimate of emissions intensity in this region (248 g CO₂e per m³ of water) agrees quite well with a bottom-up estimate produced by a previous local study of the Kansas High Plains Aquifer (231 g CO₂e per m³)²⁸ (Table S1). For the Kansas High Plains Aquifer specifically, we calculated an emissions rate of 275 g CO₂e per m³.

Because the article attempts to provide a spatial landscape of emissions, it would help to add county and state demarcation to the maps.

We have modified all the maps in order to include state boundaries. County boundaries make the color gradient difficult to see, particularly for small counties, so we have opted to not include them at this point.

Furthermore, the narrative must include a discussion of the Colorado River Basin in the accounting of groundwater vs surface water use and emissions. The proportion within the basin would be 3 (surface) to 1 (groundwater) within the basin versus majority groundwater elsewhere. Consequently, the national median may not be the best benchmark to compare results against and adding Colorado basin and non-CB could be informative.

In response to this comment, we conducted a regional analysis of the Colorado River Basin comparable to that which was conducted for the High Plains Aquifer. We have added the following text to the results to enable comparisons between the Colorado River Basin, the High Plains Aquifer area, and the national values:

Lines 210-219: The Colorado River Basin serves as a useful counterpoint. Despite its more arid climate, it has lower emissions intensities than the High Plains Aquifer region (666 kg CO₂e ha⁻¹ versus 1080 kg CO₂e ha⁻¹) because of differences in water and fuel sources. The Colorado River Basin relies heavily on surface water (72.4% of total withdrawals) instead of groundwater, and it is heavily electrified, with 89.5% of emissions on average coming from electric pumps across the 7 Colorado River Basin states. Thus, the share of national pumping emissions that occur in the Colorado River Basin (9.0%) is roughly proportionate to its share of irrigated area (7.4%). Similarly, high rates of electric pump adoption, low relative reliance on groundwater, and low electrical grid emissions intensity interact to produce relatively low emissions intensity in the Northwestern US despite high irrigation water use.

Perhaps the most interesting thought experiment in the article is the emissions projection based on electrification trends. In this regard, a one-year estimate seems insufficient to make projections. It seems appropriate to have at least two data points to project a line. Furthermore, given the identified correlation between electric pumps and groundwater use, I wonder if the increased in pumping lifts from depleting aquifers is considered for these projections. These details must be made transparent in article to provide adequate caveats to the estimates.

We developed the scenario analyses primarily to evaluate the effects of the Inflation Reduction Act on electrical grid decarbonization and subsequently on irrigation pumping emissions. For this initial analysis, we assume that the energy demand and the fuel mix remain constant over time and vary only the emissions factors for electricity use. These annual emissions factors projections were developed by the National Renewable Energy Laboratory (NREL) “using the Regional Energy Deployment System (ReEDS) model, which projects utility-scale electricity

sector evolution for the contiguous United States using a system-wide, least-cost approach subject to policy and operational constraints,” as per the NREL 2022 Standard Scenarios Technical Report (reference 36). The scenarios include simulations with and without the IRA, and with several variations representing future policy changes such as the extension of key tax credits. The first decarbonization analysis reflects only changes in the grid emissions factors as projected by the NREL ReEDS model, informed by historical trends, policy, and infrastructural constraints.

For an additional analysis, we added an arbitrary rate of pump electrification (5% of remaining non-electric fuels per year) to provide a reference point for the impact of hypothetical fuel switching on emissions. As you point out, however, we are not attempting to project likely changes in electrification, nor are we considering any other variables that would influence pump energy demand such as changes to groundwater levels or crop water demand. Rather than providing projections of likely future irrigation emissions, this analysis was intended to provide a simplified representation of the impacts of two key levers of pumping decarbonization.

We have added the following text to 1) clarify the intent of the analysis and associated caveats, and 2) discuss the value of further scenario analysis for comprehensively evaluating likely future emissions:

Lines 351-353: This rate does not attempt to reflect historical trends, but rather serves as a benchmark to exemplify decarbonization potential associated with electrification.

Lines 370-375: Additionally, scenario analyses that integrate the effects of pump electrification and grid decarbonization with variables that will influence pump energy demand would be useful to clarify future projections of irrigation-related emissions. For instance, declining groundwater levels, migration of irrigation and crop types, changes to surface versus groundwater reliance, and changes to irrigation water demand due to climate change will all affect total pump energy requirements.

Doing a quick search on the topic and taking a couple of first-page articles, it appears that more discussion on the current state of knowledge is required. Rothausen and Conway (2011) provide estimates of irrigation sector energy use or emissions from various countries which would be worthwhile overviewing for benchmark to your results. Sapkota et al. (2020) provide micro-estimates of irrigation GHG emissions (field-level) which should be used to benchmark the deduced macro-estimates in your article. I’m no longer convinced that being the only ones providing estimates is sufficient for publication without discussion as to why these estimates haven’t been presented before (difficulties, biases, etc.) or a good overview of the current state of knowledge on the topic, from whichever region.

- Rothausen, S.G. and Conway, D., 2011. Greenhouse-gas emissions from energy use in the water sector. Nature Climate Change, 1(4), pp.210-219.

- Sapkota, A., Haghverdi, A., Avila, C.C. and Ying, S.C., 2020. Irrigation and greenhouse gas

emissions: a review of field-based studies. Soil Systems, 4(2), p.20.
There must be others.

We agree that additional context from the existing literature was needed for our estimates and feel that the changes implemented in response to this and similar comments from other reviewers have substantially strengthened the manuscript. We have added a table of published estimates of total energy use, total GHG emissions, and per-hectare and per-volume emissions and energy intensities for irrigation pumping, as available (Table S1). This table leverages selected international and regional studies which provide estimates of energy use and/or emissions based on primary data on well depth, fuel type, and pumping volumes, or direct data on energy use. Because Sapkota et al. 2020 deals explicitly with biogeochemical impacts rather than energy use, we do not think that it provides a point of comparison for our energy use estimates. We do leverage several of the studies referenced in the review provided by Rothausen & Conway (2011) but choose to cite the primary studies rather than their review. We have also added the following text situating our results within the context of previous estimates:

Lines 93-103: This corresponds to an estimated 156 PJ of total energy use for on-farm irrigation pumping. Previous studies have estimated on-farm irrigation pump energy use at 158 PJ nationally³⁰ and 136 PJ for electricity use in the Western USA²⁹, in close agreement with our estimates. Table S1 provides energy and emissions intensity estimates from selected regional and international studies, demonstrating substantial variability in estimated intensities. On a per-hectare basis, for instance, energy intensity estimates have ranged from 6,687 MJ ha⁻¹ (our study) up to 43,412 MJ ha⁻¹ (from a study of groundwater irrigation in Pakistan), while emissions intensity estimates have varied from 0.54 tonnes CO₂e ha⁻¹ (our study) up to 1.27 tonnes CO₂e ha⁻¹ (from a study of groundwater irrigation in India). Previous studies have often focused on groundwater pumping, which is associated with higher energy use and subsequently higher emissions.

REVIEWERS' COMMENTS

Reviewer #2 (Remarks to the Author):

The authors did an excellent job with the responses to all reviewer's comments. The selected revisions and reanalyses that they undertook have improved the manuscript and its presentation substantially. I look forward to seeing this in print and using it for future work!

Reviewer #3 (Remarks to the Author):

Thank you for working through each of the comments. I am satisfied that you have done the best effort possible to address them.